# Melanocortin-5 Receptor: Pharmacology and Its Regulation of Energy Metabolism

**DOI:** 10.3390/ijms23158727

**Published:** 2022-08-05

**Authors:** Li-Qin Ji, Ye Hong, Ya-Xiong Tao

**Affiliations:** 1Shandong Provincial Key Laboratory of Animal Cell and Developmental Biology, School of Life Sciences, Shandong University, Qingdao 266237, China; 2Department of Anatomy, Physiology and Pharmacology, College of Veterinary Medicine, Auburn University, Auburn, AL 36849, USA

**Keywords:** melanocortin-5 receptor, pharmacology, melanocortin-2 receptor accessory protein, energy metabolism, signaling pathway

## Abstract

As the most recent melanocortin receptor (MCR) identified, melanocortin-5 receptor (MC5R) has unique tissue expression patterns, pharmacological properties, and physiological functions. Different from the other four MCR subtypes, MC5R is widely distributed in both the central nervous system and peripheral tissues and is associated with multiple functions. MC5R in sebaceous and preputial glands regulates lipid production and sexual behavior, respectively. MC5R expressed in immune cells is involved in immunomodulation. Among the five MCRs, MC5R is the predominant subtype expressed in skeletal muscle and white adipose tissue, tissues critical for energy metabolism. Activated MC5R triggers lipid mobilization in adipocytes and glucose uptake in skeletal muscle. Therefore, MC5R is a potential target for treating patients with obesity and diabetes mellitus. Melanocortin-2 receptor accessory proteins can modulate the cell surface expression, dimerization, and pharmacology of MC5R. This minireview summarizes the molecular and pharmacological properties of MC5R and highlights the progress made on MC5R in energy metabolism. We poInt. out knowledge gaps that need to be explored in the future.

## 1. Introduction

Melanocortin receptors (MCRs), members of Family A (rhodopsin-like) G protein-coupled receptors (GPCRs), consist of five members (MC1R to MC5R) with diverse biological functions [1,2]. MC1R is involved in pigmentation and inflammation [3,4,5,6,7]. MC2R, exclusively found in the adrenal gland and activated by adrenocorticotropic hormone (ACTH), regulates steroid production and cell proliferation [8,9,10,11]. Centrally expressed MC3R and MC4R have essential non-redundant functions in energy homeostasis [12,13,14,15,16,17,18]. Cloning of the five MCRs from 1992 to 1995 started a new research phase due to the specific pharmacological properties of these five MCRs and their therapeutic potential in treating diverse diseases [2,5,19,20].

The melanocortin system plays important roles in regulating energy homeostasis. Neural MC3R and MC4R, highly expressed in the hypothalamus, can sense and integrate external stimuli (humoral and nutrient cues), including leptin, insulin, ghrelin, serotonin, orexin, and glucose, regulating energy homeostasis in different ways [21,22,23,24,25]. MC3R regulates feed efficiency [26,27,28], feeding rhythm [29,30,31], and energy expenditure [32], whereas MC4R regulates both food intake and energy expenditure [13,25]. Selectively reactivating MC4R expression in specific neurons showed that the MC4R expressed in the paraventricular nucleus of the hypothalamus and amygdala is involved in the regulation of food intake, whereas MC4R expressed in other neurons is involved in controlling energy expenditure [33]. For general reviews on MC3R and MC4R, the reader is referred to several review articles [13,14,34,35,36].

MC5R, widely expressed in central and peripheral tissues, has multiple physiological functions (Figure 1). In the brain, MC5R is involved in the stress response [37], cognitive function [38], and fetal brain development [39]. MC5R in the perifornical lateral hypothalamus might mediate physical activity in lean rats [40]. In peripheral tissues, MC5R is involved in exocrine and endocrine gland secretion [41,42], defense behavior [43,44], thermoregulation [41], inflammation [45,46], and immune response [47,48,49,50]. MC5R regulates energy metabolism in the liver, adipose tissue, and skeletal muscle of various species, such as humans [51,52,53,54], mice [51,55,56,57,58,59,60], chickens [61], and sea bass [62]. MC5R primarily regulates energy metabolism via adipocyte lipolysis and re-esterification, fatty acid oxidation, and glucose uptake [51,52,53,54].

In contrast to MC3R and MC4R, studies on MC5R are very limited [41,42,43,44,45,47,48,49,50,63,64,65,66]. Moreover, the role of MC5R in energy metabolism has been rarely investigated. Herein, we summarize the molecular characteristics and pharmacology of MC5R, including the signaling pathways, as well as physiological functions, especially in energy metabolism, by comparing it with MC3R and MC4R.

## 2. Molecular Characteristics of MC5R

As the most recent member of MCRs to be cloned, MC5R was identified from rodent and human genomic DNA in 1994 and 1995 [67,68]. The intronless *MC5R* is located on chromosome 18p11.21, encoding 325 amino acids in humans [69]. MC5R consists of seven putative hydrophobic transmembrane domains (TMDs) linked by alternating extracellular and intracellular loops (ECLs and ICLs, respectively), with an extracellular N-terminus and intracellular C-terminus (Figure 2). The amino acid sequences of MC5Rs in vertebrates are highly conserved at TMDs, while N-terminal extracellular domains display the lowest identity (Figure 3).

There is disagreement on the evolutionary relationship and the origin of MC5R. Genomic analysis shows that *MC5R* is consistently adjacent to *MC2R* in the opposite direction on the same chromosome (Figure 4). The conserved synteny between *MC2R* and *MC5R* in many species indicates that they might have evolved from a common ancestor by local duplication. This event could date back to the ancestral gnathostome since elasmobranchs have both *mc2r* and *mc5r* [70,71]. However, another view posits that MC5R originated from a local duplication of MC4R, and then the *MC5R* locus was transferred next to the *MC2R* locus [72]. This discrepancy may be attributed to the different evolutionary methods used [73].

To date, *MC5R* genes have been cloned from multiple species of vertebrates, including fish, amphibians, birds, and mammals. There are two *mc5r* subtypes in zebrafish, *mc5ra* and *mc5rb*, resulting from gene duplication during evolution [74,75]. However, *MC5R* is absent or inactivated in some placental lineages owing to their completely lost or degenerative sebaceous glands, such as Cetacea, West Indian manatee, African elephant, and white rhinoceros [76]. The differential loss of *MC5R* in whales and manatees was suggested to be the result of convergent evolution in the marine environment [77]. 

In contrast to neural MC3R and MC4R, MC5R is widely expressed in central and peripheral tissues, such as the brain, exocrine glands, skin, adipose tissue, skeletal muscle, kidney, liver, and other tissues (Table 1 and Figure 5). In different species, MC5R shows divergent expression patterns. For example, *Mc5r* mRNA is low in the central nervous system but abundant in a variety of peripheral tissues in mice and rats [42,78]. Detailed profiling of *Mc5r* in mice showed that it is highly expressed in the whole eye, skeletal muscle, urinary bladder, and skin and moderately expressed in the vena cava, adipose tissue (including both brown and white adipose tissues), and the central nervous system [79]. However, *mc5r* cloned in fishes showed high levels of *mc5r* transcripts in the brain and pituitary in some fishes [37,80,81,82,83,84]. The wide distribution of MC5R in multiple tissues might contribute to its diverse functions.

There are two reports of human *MC5R* mRNA expression (Figure 5A and Table 1) [51,85]. An earlier study reported *MC5R* mRNA expression in the brain, pancreas, lung, heart, testes, and adipose tissue [51], whereas *MC5R* mRNA in the Human Protein Atlas database shows abundant expression in the epididymis, esophagus, and thymus, as well as low expression in the brain, retina, skin, and others [85]. Further studies using multiple sensitive techniques, such as NanoString nCounter Technology [86], are needed to further clarify the tissue distribution of human *MC5R*.

**Table 1 ijms-23-08727-t001:** The distribution of MC5R in different species.

Species	MC5R Expression in Different Tissues	Techniques
Human [51]	Present in brain, pancreas, lung, heart, testes, and fat tissues	RT-PCR
Mouse [41,78]	Abundant in the Harderian, lacrimal, and preputial glands; moderate in muscle and skin; low levels in adipose, spinal cord, and brain; absent in spleen, kidney, liver, heart, lung, and gonad	In situ hybridization
Rat [42]	Abundant in lacrimal, preputial, and Harderian glands; low levels in adrenal glands, pancreas, esophagus, and thymus; absent in thyroid gland, seminal vesicle, spleen, liver, and skeletal muscle	Western blot, In situ hybridization
Chicken [87]	Present in brain, kidney, liver, adrenals, ovary, testis, uropygial gland, and adipose tissue; absent in heart, spleen, and skeletal muscle	RT-PCR
Zebrafish [74]	Present in ovary, brain, gastrointestinal tract, and eye (*mc5ra*); present in ovary, brain, gastrointestinal tract, eye, and heart (*mc5rb*)	RT-PCR
Barfin flounder [88]	Present in pituitary, brain, eyeball, gill, atrium, ventricle, liver, head kidney, kidney, spleen, stomach, intestine, white muscle, inclinator muscle, testis, ovary, and skin	RT–PCR
Sea bass [62]	Present in retina, brain, liver, spleen, gill, testis, and dorsal skin; low levels in the pituitary, posterior kidney, fat tissue, intestine, red muscle, and ovary	RT–PCR
Goldfish [80]	Present in the kidney, spleen, skin, retina, and brain; low levels in the intestine, fat, muscle, gill, pituitary, and ovary	RT–PCR, Southern blot
Common carp [81]	Present in brain, skin, kidney, and pituitary; absent in thymus, spleen, head kidney, gut, gill, liver, heart, and muscle	RT–PCR
Blunt snout bream [37]	Present in brain, eyes, skin, testis, ovary, and gill; low levels in the muscle, intestine, kidney, head kidney, spleen, and liver	RT–PCR
Horn shark [71]	Present in brain, pituitary, skin, and liver	RT–PCR
Stingray [89]	Present in hypothalamus and inter-renal tissues	RT–PCR
Elephant shark [10]	Present in hypothalamus, pituitary, brain, and kidney	RT–PCR

## 3. Pharmacology of MC5R

### MC5R Ligands

The natural ligands for MCRs are melanocortins as agonists and two endogenous antagonists, namely, agouti (or agouti-signaling protein, ASIP) and agouti-related protein (AgRP). Melanocortins, including ACTH and α-, β-, and γ-melanocyte-stimulating hormones (α-, β-, and γ-MSHs), are formed by post-translational processing of the precursor, proopiomelanocortin (POMC) [1,2,5,90]. These products are mainly expressed in the hypothalamus and pituitary as well as in the skin [91,92,93]. α- and β-MSHs are part of ACTH; therefore, they share the same core sequence, the pharmacophore, His-Phe-Arg-Trp, which is necessary for receptor binding and activation [94,95]. Endogenous melanocortins are able to nonspecifically activate MC5R in many species, from fish to mammals [37,62,67,74,84,96,97,98]. Generally, MC5R displays the highest affinity to α-MSH but the lowest to γ-MSH in mice [67], humans [84,96], and fishes, such as stingray [97], zebrafish [74], blunt snout bream [37], and ricefield eel [84].

To obtain more potent ligands, several labs have developed synthetic agonists for MC5R. Some synthetic ligands display higher potency for MC5R than endogenous agonists, such as [Nle^4^-D-Phe^7^]-α-MSH (a synthetic superpotent analog of α-MSH), melanotan II (MTII), SHU9119 (MTII and SHU9119 are potent cyclic derivatives of α-MSH), and HS014 (reviewed in [99]). However, these synthetic ligands can also effectively activate (or antagonize, as in the case of SHU9119) the other MCR subtypes, suggesting that they do not exhibit good selectivity for MC5R. Subsequently, agonists highly specific to MC5R were developed, including PG-901, PG-911, OBP-MTII (Oic^6^, D-4,4′-Bip^7^, Pip^8^-MTII), and others [99,100,101]. 

ASIP and AgRP are endogenous antagonists in the melanocortin system [102,103,104,105,106]. The modification of pharmacophores (Arg-Phe-Phe-Asn-Ala-Phe) on exposed β-hairpin loops of AgRP or ASIP can improve the antagonist potency or cause a functional change from an antagonist to an inverse agonist for MC5R. For example, c[Pro-Arg-Phe-Phe-Asn-*Val*-Phe-_D_Pro] and c[Pro-Arg-*Tyr*-Phe-Asn-Ala-Phe-_D_Pro] were found to more efficiently antagonize MC5R [107]. The design of highly potent and selective ligands is essential for developing molecular probes to identify new functions of MC5R.

As a typical GPCR, MC5R binding to agonists activates the Gα subunit by the exchange of GDP for GTP and the dissociation of the Gα subunit from the Gβγ dimer and from the receptor. Activated MC5R can be coupled to the cAMP pathway via Gαs and the Ca^2+^ pathway via Gαq [108]. cAMP triggers downstream events such as lipolysis and inflammation [109]. Moreover, MC5R can activate some pathways independent of cAMP and Ca^2+^. For example, MC5R triggers the PI3K-ERK1/2 pathway, which can further mediate downstream pathways in fatty acid re-esterification [110], cellular proliferation/differentiation, and immune responses [111].

## 4. The Effect of MRAPs on MC5R Pharmacology

Melanocortin-2 receptor (MC2R) accessory protein (MRAP) was initially discovered as an essential partner for MC2R by assisting in MC2R trafficking from the endoplasmic reticulum to the cell surface [112,113,114]. MRAP2, a subsequently discovered homolog of MRAP, exhibits similar functions to MRAP in adrenal differentiation and proliferation [115]. Both MRAPs show wide tissue distribution in the central nervous system, especially in the hypothalamus, and peripheral tissues, including the pituitary, adrenal glands, testis, adipose tissue, ovary, and digestive tract [112,116,117,118] (Figure 5B,C).

Subsequent studies showed that MRAPs can also regulate MC5R trafficking and pharmacology in many species (Table 2). MRAPs disrupt MC5R dimerization in humans and zebrafish [75,119] and regulate MC5R trafficking to the plasma membrane. For example, MRAPs inhibit MC5R trafficking to the plasma membrane in humans and zebrafish [75,116,119], whereas they increase MC5R trafficking in gar [120]. However, MRAPs may modulate MC5R pharmacology independent of receptor trafficking in some species, such as mouse, elephant shark, whale shark, and ricefield eel [10,75,84,121] (Table 2).

Co-expression of *MC5R* and *MRAP* in the same cells or tissues is the rationale for their interaction. The Human Protein Atlas database showed that human *MC5R* mRNA and *MRAP1*/*MRAP2* are expressed in the same tissues, including the brain, esophagus, testis, epididymis, skin, and thymus [85] (Figure 5). Similarly, mouse *Mc5r* and *Mrap2* mRNA are expressed in the brain, skin, muscle, and adipose [78,118]. Future research should systematically investigate the interaction of MC5R and MRAPs in the same cells in these tissues.

## 5. Functions of MC5R in Energy Metabolism

Knockout mouse models have elucidated the functions of MCRs. *Mc4r*^−/−^ mice exhibit severe phenotypes in energy homeostasis, including hyperphagia, mature-onset obesity, increased linear growth, hyperinsulinemia, and hyperglycemia [123]. Unlike the hyperphagia and severe obesity phenotype in *Mc4r*^−/−^ mice, homozygous *Mc3r* knockout mice exhibit a mild phenotype, characterized by moderate obesity and no hyperphagia but elevated fat mass and reduced lean mass [26,27] (Table 3).

No obvious deficiency in appearance, behavior, growth, or reproduction was observed in *Mc5r* knockout mice. Other parameters associated with metabolic homeostasis in *Mc5r*-deficient mice are indistinguishable from those of their wild-type littermates, including muscle mass, adipose mass, and blood glucose and insulin levels. However, *Mc5r* knockout mice are deficient in the secretion of multiple exocrine glands, including Harderian porphyrin production and lacrimal protein secretion [41]. In *Mc5r* knockout mice, total acetone-extractable lipids from hair are decreased by 15–20%, which leads to defective water repulsion and thermoregulation. Another study on glucose metabolism found that α-MSH-activated MC5R increases thermogenesis, glucose uptake, and whole-body glucose clearance in skeletal muscles in wild-type mice, whereas these actions are inhibited in *Mc5r* knockout mice [124].

## 6. MC5R Regulates Lipolysis and Re-Esterification

Obesity is characterized by the expansion of adipose tissue caused by triacylglycerol (TAG) accumulation in adipocytes [135]. The adipocytes in white adipose tissue are a site of fat storage, mediated by TAG synthesis (lipogenesis) and degradation (lipolysis). Lipolysis is a biochemical process involving the breakdown of triglycerides and the release of non-esterified fatty acids and glycerol [135,136,137,138,139]. Lipolysis is catalyzed by three major enzymes: hormone-sensitive lipase, adipose triglyceride lipase, and monoacylglycerol lipase [135,137].

Despite the lack of the dramatic metabolic phenotype of *Mc5r*^−/−^ mice, *Mc5r* has been shown to be expressed in mouse adipocytes and differentiated 3T3-L1 mouse adipocyte cells [140]. In 3T3-L1 cells, α-MSH-stimulated MC5R activates hormone-sensitive lipase and perilipin-1, inducing lipolysis by activating the cAMP/PKA signaling pathway, whereas MC5R prevents triglyceride synthesis by inhibiting the function of acetyl-CoA carboxylase (ACC), an important enzyme in the lipogenic process [110,141] (Figure 6). In addition, MC5R inhibits re-esterification by blocking the recycling of non-esterified fatty acids into triglycerides via ERK1/2 signaling in mouse 3T3-L1 adipocytes [110]. Moreover, it was found that the lipolytic function of MC5R is dependent on noradrenalin released from postsynaptic nerve fibers innervating the adipose tissue in humans [54]. In addition, MC5R in 3T3-L1 adipocytes can inhibit leptin secretion, supporting the possibility that MC5R indirectly regulates food intake and energy expenditure by leptin–melanocortin pathways [142]. The in vivo physiological relevance of these observations remains to be established since the endogenous level of α-MSH in adipose tissue might not be sufficient to fully activate MC5R [143]. The expression of MCRs in human adipocytes is also lower or absent in humans, different from rodents [143]. The function of MC5R in lipolysis has also been identified in chicken and sea bass [61,62,144]. 

*MC5R* mutations in Quebec families and Finns exhibit significant linkage or association with the obesity phenotype [51,53]. However, detailed studies on the mutations identified are insufficient to prove a causal relationship between the mutation and human obesity. As shown in Figure 2, numerous additional *MC5R* mutations have been identified by recent extensive genomic studies. Whether these *MC5R* mutations lead to defective mutant receptors and the exact molecular defects remain to be studied. The correlation of a molecular defect with a phenotype will be necessary to convincingly demonstrate the clinical implications of these mutations in human diseases. We have performed extensive functional studies on naturally occurring mutations in the related *MC3R* and *MC4R* [25,145,146,147,148,149,150,151,152,153,154,155,156,157,158]. Importantly, these studies identified potential strategies to correct these mutations, especially pharmacological chaperones for correcting misfolded mutant receptors [13,151,159,160,161]. Similar studies need to be conducted with naturally occurring mutations in *MC5R*. 

## 7. MC5R Regulates Fatty Acid Oxidation

In humans, skeletal muscle, accounting for more than 70% of total glucose disposal in the body, is an important tissue in determining whole-body energy expenditure [162]. Long-chain fatty acids, mainly derived from adipocyte lipolysis, are transported into skeletal muscle, where it is partly oxidized to provide energy. Fatty acid oxidation (FAO) in skeletal muscle occurs in the mitochondria, which is promoted by the actions of carnitine palmitoyltransferase-1 (CPT-1). CPT-1 activity is negatively mediated by malonyl-CoA, which is synthesized from cytosolic acetyl-CoA through a reaction catalyzed by ACC [56,163,164]. In exercising skeletal muscle, activation of 5′-AMP-activated protein kinase (AMPK) facilitates glucose transport and FAO through the inhibition of ACC, which leads to a decrease in malonyl-CoA content and an increase in CPT-1 activity [56,164].

Among all MCRs, MC5R is the predominant subtype expressed in skeletal muscle, suggesting potential important functions of this receptor in skeletal muscle [41,56,78,87]. α-MSH-activated MC5R enhances FAO in mouse muscle cells and C2C12 myoblast cells. Activated MC5R triggers the cAMP-PKA-AMPK pathway, followed by ACC phosphorylation, which suppresses ACC activity but increases CPT-1 activity, leading to improved FAO [56] (Figure 6). In addition, C/EBPβ binds to the promoter region of MC5R and acts as a negative transcription regulator. α-MSH can reduce the interaction of C/EBPβ with MC5R to enhance FAO in white and brown adipocytes [59].

## 8. MC5R Regulates Glucose Homeostasis

Glucose uptake is a process in which glucose in the blood is transferred into the cell via multiple glucose transporters (GLUTs). In skeletal muscle, three GLUTs are involved in glucose uptake: GLUT4, GLUT1, and GLUT3 (expressed in fetal and neonatal muscle only). GLUT1 is constitutively expressed on the plasma membrane, whereas GLUT4 is transported to the cell surface by intracellular vesicles in response to stimuli [165]. AMPK can regulate glucose uptake via phosphorylation of two downstream targets, AS160 and TBC1 domain family member 1 (TBC1D1) [166]. Phosphorylated AS160 and TBC1D1 were demonstrated to promote GLUT4 translocation in skeletal muscle, adipose tissue, and other peripheral tissues [167,168]. Skeletal muscle accounts for 15–20% of total glucose disposal in the basal state, and it takes up approximately 80% of glucose after a meal [165,169].

Single nucleotide polymorphisms in *MC5R* are associated with type 2 diabetes and obesity in Finns, suggesting that MC5R might be involved in glucose disposal in humans [53]. Further study found that α-MSH stimulates glucose uptake and induces the phosphorylation of TBC1D1, which is not regulated by upstream PKA and AMPK in mouse soleus muscles. Moreover, α-MSH-mediated glucose uptake is not exerted by GLUT4 [60] (Figure 6).

Pituitary and extra-pituitary cells, including keratinocytes, monocytes, astrocytes, and gastrointestinal cells, can produce peripheral α-MSH [124,170,171]. The pituitary gland, which expresses *POMC*, is composed of an anterior lobe, an intermediate lobe, and a neural lobe. The anterior lobe in humans and the intermediate lobe in most mammals are the dominant origins of circulating α-MSH [172], accounting for approximately 70% of blood α-MSH in higher mammals [124,170,171]. Pituitary POMC cells can sense plasma glucose fluctuations, which, in turn, stimulates the secretion of circulating α-MSH in humans, mice, and monkeys [124].

Experiments with sheep and *Mc5r* knockout mice found that physiological levels of circulating α-MSH increase thermogenesis, glucose tolerance, and muscle glucose uptake in skeletal muscle via increased glycolysis and anaerobic respiration to produce ATP and lactic acid [124]. Moreover, these actions of α-MSH are dependent on the MC5R-cAMP-PKA signal transduction pathway in the soleus and gastrocnemius muscles of lean animals, whereas the effect of α-MSH on glucose uptake is abolished in *Mc5r* knockout mice [124]. Further study found that α-MSH stimulates glucose uptake and induces the phosphorylation of TBC1D1, which is not regulated by upstream PKA and AMPK in mouse soleus muscles. Moreover, α-MSH-mediated glucose uptake is not exerted by GLUT4 (Figure 6). Since high levels of both MC4R and MC5R are detected in mouse soleus muscle, the role of MC4R in glucose uptake is not clear [60]. 

## 9. Future Perspectives

Compared with the other four MCRs, studies on the structure–function relationships of MC5R are very limited. Crystal structures have been recently described for MC1R and MC4R. Elucidation of the crystal structure of MC5R will facilitate the in silico design of novel ligands for MC5R, especially small molecules. The development of subtype-selective ligands is of special interest in that these ligands can be used to study the physiology of MC5R in species other than rodents.

Since MC5R is widely expressed, it is likely to have multiple functions in different tissues. Preliminary clinical studies indicated that *MC5R* is associated with obesity, and recent genetic studies have identified many novel mutations in *MC5R*. However, the functional and clinical relevance of these mutations remain to be investigated.

In vitro studies showed that MRAPs can regulate the pharmacology of MC5R in HEK293 or CHO cells, indicating the potential of MRAPs to regulate the function of MC5R. Co-expression of *MC5R* and *MRAP1/MRAP2* in different tissues, especially in the same cells of these tissues, and the functional regulation of MC5R by MRAPs in a physiological environment need to be studied. 

The different physiological functions of MC5R have been mostly reported by a single lab. Confirmation by independent labs and further extension of physiological studies, including the use of tissue-specific knockout and receptor subtype-selective ligands, are needed. Importantly, the pharmacological properties of the tools to be used also need to be independently confirmed, rather than just relying on previous publications. Tissue-specific knockout of *Mc5r* will likely yield clues to the functions of MC5R in different tissues. Since energy homeostasis can be affected by multiple environmental stimuli, such as glucose intake, high-fat diet, fasting, and feeding rhythm, it would be beneficial to investigate the phenotype of *Mc5r* knockout mice upon these challenges.

## Figures and Tables

**Figure 1 ijms-23-08727-f001:**
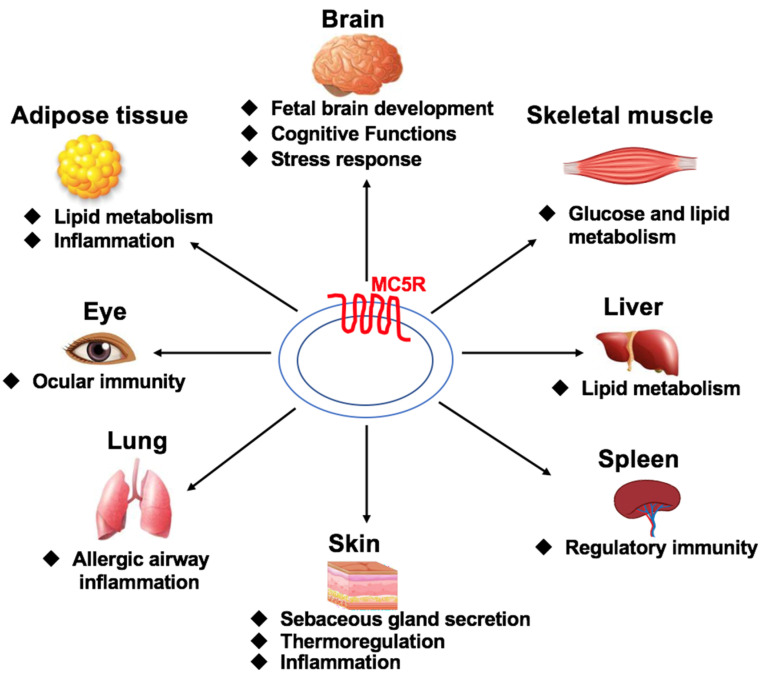
Multiple functions of MC5R in various tissues.

**Figure 2 ijms-23-08727-f002:**
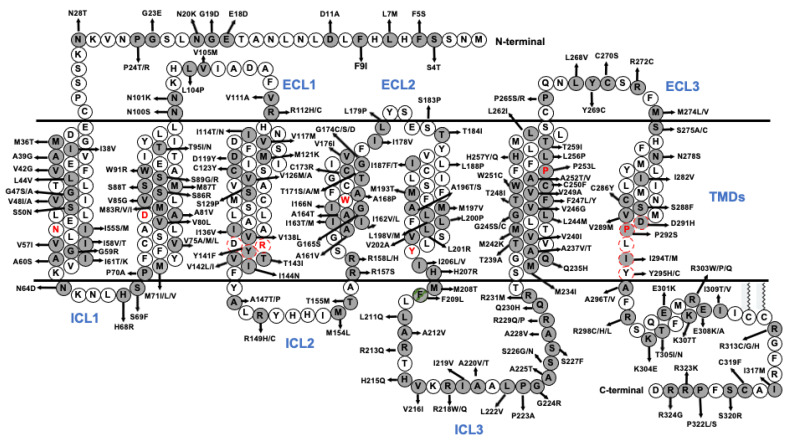
Naturally occurring human *MC5R* mutations recorded in gnomAD database v2.1.1 (https://gnomad.broadinstitute.org/ (accessed on 13 April 2022)). The circles with gray background are missense and nonsense mutations/polymorphisms. Frameshift mutations are not shown here. The polymorphism (F209L) is labeled with a circle filled with red dashed lines. The most conserved residues in transmembrane domains (TMDs) are denoted in red font. DRY and DPxxY motifs are labeled in dashed line circles. MC5R secondary structures with extracellular loops (ECLs), transmembrane domains (TMDs), and intracellular (ICLs) loops are denoted in blue font.

**Figure 3 ijms-23-08727-f003:**
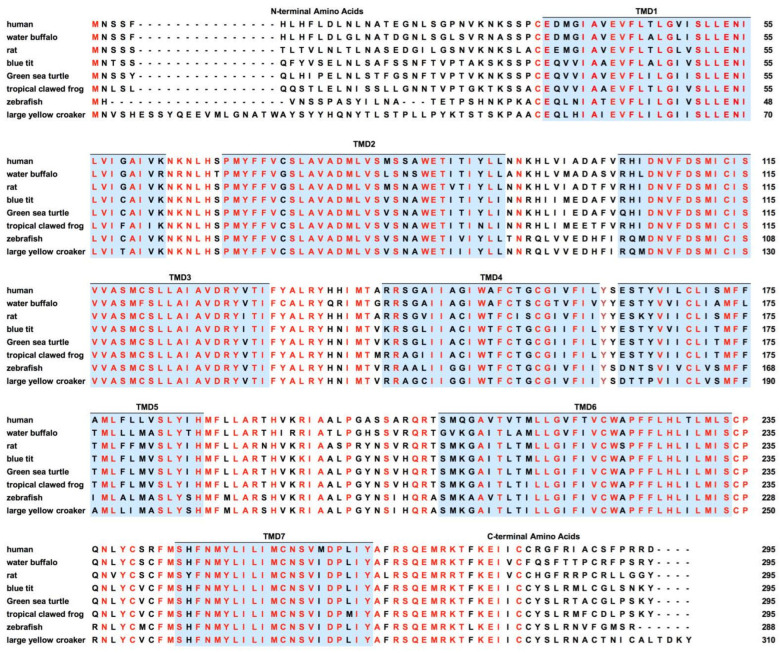
Sequence alignment of multiple MC5Rs. The transmembrane (TM) regions are represented by blue shadow and are numbered 1–7. The 100% identical residues are indicated in red. MC5Rs: *Homo sapiens* (human, NP_005904.1), *Mus musculus* (mouse, NP_038624.3), *Bubalus bubalis* (water buffalo, XP_025129279.1), *Cyanistes caeruleus* (blue tit, XP_023777141.1), *Chelonia mydas* (green sea turtle, XP_007063924.1), *Xenopus tropicalis* (tropical clawed frog, NP_001096392.1), *Danio rerio* (zebrafish, NP_775386.1), and *Larimichthys crocea* (large yellow croaker, XP_010746135.1).

**Figure 4 ijms-23-08727-f004:**
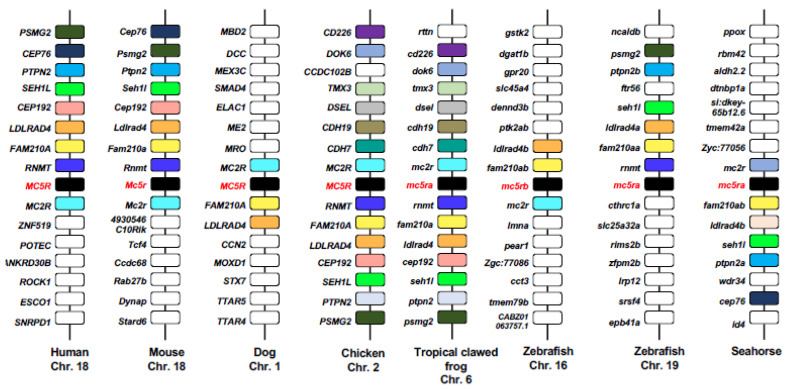
Comparative synteny analysis of *MC5R*. Chromosomal location and adjacent genes of *MC5R* are shown in different species. Genes with conserved synteny between at least two species are shown in the boxes with the same color (except the white box).

**Figure 5 ijms-23-08727-f005:**
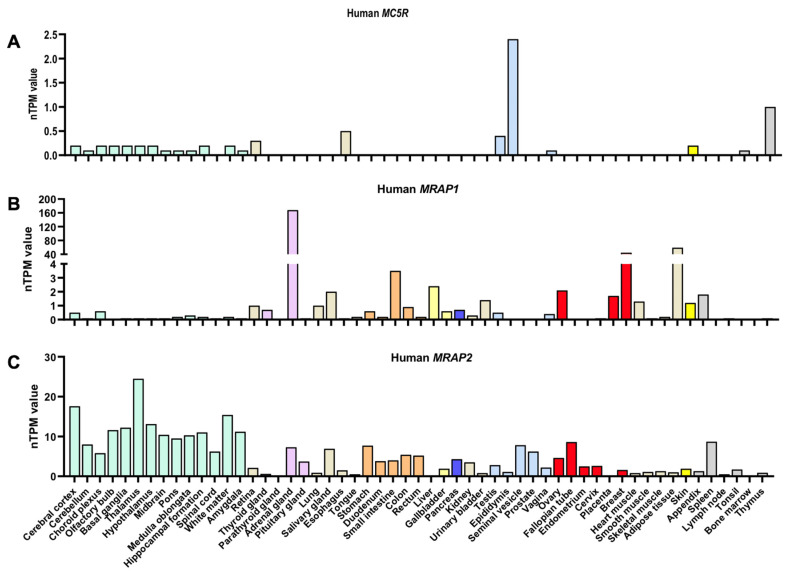
Human *MC5R* (**A**), *MRAP1* (**B**), and *MRAP2* (**C**) mRNA expression in various tissues, based on https://www.proteinatlas.org/ (accessed on 14 July 2022) [85]. nTPM indicates normalized protein-coding transcripts per million. Color coding is based on tissue groups with functional features in common.

**Figure 6 ijms-23-08727-f006:**
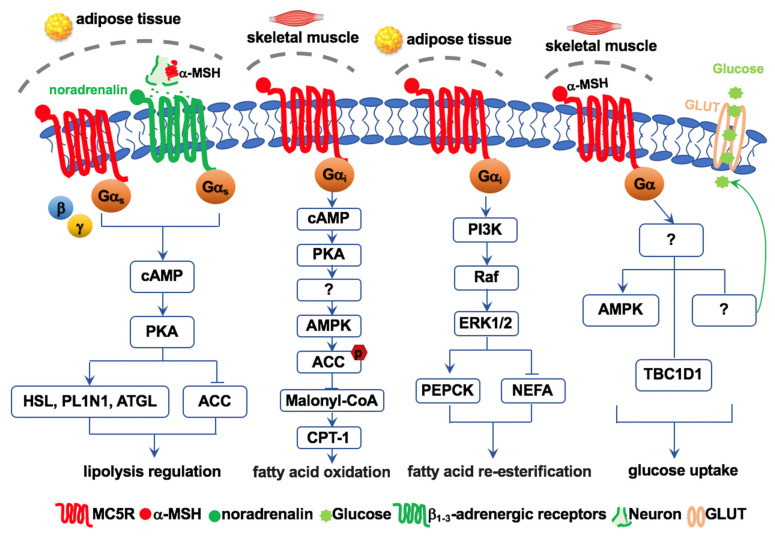
Schematic diagram of MC5R signaling pathways in lipid and glucose metabolism.

**Table 2 ijms-23-08727-t002:** The effect of MRAPs on MC5R in various species.

Species	MRAPs	Effect of MRAPs on MC5R-Related Parameters		Cell Types
		MC5R Traffic to PM	MC5R Pharmacology	
Human [116,119]	MRAP1, MRAP2	Inhibition *	Inhibit its efficacy for NDP-MSH *	CHO HEK293T
Zebrafish [75]	MRAP2a	Inhibition	Inhibits the efficacy of both MC5Ra and MC5Rb with α-MSH and SHU9119	CHO HEK293T
	MRAP2b	NS	Inhibits MC5Ra but increases MC5Rb efficacy with α-MSH and SHU9119	
Mouse [75]	MRAP2	NS	Inhibits efficacy with α-MSH and SHU9119	CHO HEK293T
	MRAP1	—	—	—
Elephant shark [10]	MRAP1	NS	Increases sensitivity to ACTH but not Des-Acetyl-α-MSH	CHO
	MRAP2	NS	NS	
Chicken [122]	MRAP1	—	Increases sensitivity to ACTH	CHO
	MRAP2	—	No effect on responding to ACTH	
Gar [120]	MRAP1	Increase	Increases efficacy with NDP-MSH	CHO
	MRAP2	NS	Increases efficacy with ACTH	
Whale shark [121]	MRAP1, MRAP2	NS *	Increase sensitivity to ACTH but not des-acetyl-α-MSH *	CHO
Ricefield eel [84]	MRAP2X1	NS	Increases maximal binding and inhibits efficacy with α-MSH and ACTH *; no influence on binding affinity to ACTH or α-MSH	HEK293T
	MRAP2X2	NS	Decreases binding affinity to ACTH but not a-MSH	
Rainbow trout [83]	MRAP2	NS	Increases sensitivity to ACTH	CHO
	MRAP	—	—	—

PM, plasma membrane; * indicates both MRAP subtypes have the same influence; NS indicates the MRAP subtype has no significant effect on the parameter; — indicates data not available.

**Table 3 ijms-23-08727-t003:** Functions of MC5R, MC4R, and MC3R in regulation of energy homeostasis.

	MC3R	MC4R	MC5R
Energy-regulating tissues	Hypothalamus [22]	Hypothalamus, adipose, and skeletal tissue [13,26,27]	Liver, adipose, and skeletal tissue [53,54,62,110,124]
Feeding behavior	Feed efficiency, feeding rhythm, and energy expenditure [26,27,28,29,30]	Food intake and energy expenditure [13,25,125]	No report
Phenotype in knockout mouse	Moderate obesity, no hyperphagia, increased fat mass, and decreased lean mass [123,126]	severe obesity, hyperphagia, and hyperinsulinemia [13,27,123,127]	No visible phenotype, deficiency in exocrine gland secretion, and decreased glucose tolerance [41,124]
Lipid homeostasis	Triglyceride accumulation, lipolysis, and fatty acid oxidation [14,128,129]	Triglyceride synthesis, lipid mobilization, and fat accumulation [129,130,131]	Lipolysis, fatty acid oxidation, and fatty acid re-esterification [53,54,62,110]
Glucose homeostasis	Glucose uptake [14,132,133]	Glucose reabsorption, hyperglycemia, and hepatic glucose production [13,16,134]	Glucose uptake [124]

## Data Availability

Not applicable.

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
