# Peer review of "Melanocortin-5 Receptor: Pharmacology and Its Regulation of Energy Metabolism"

_ijms, 2022, doi:10.3390/ijms23158727_

Round 1

Reviewer 1 Report

In the following review, the authors detail MC5R and its potential role in energy homeostasis. Overall, the first few sections are well written and give a good description of the distribution and pharmacology of MC5R. However, the impact of MC5R on energy homeostasis is superficial, and many statements made already have been examined or performed. It is suggested that the authors re-work that section to incorporate more previous findings and lead to better anaylsis of the current literature and proposal of future studies.

1) For example, the authors state in line 233 of pg 9 - " Further study using the Mc5r-/- model or selective ligands for MC5R can help to elucidate the mechanisms of MC5R involvement in glucose uptake in skeletal muscle." However, this is exactly what citation 113 did - they utilized knockout mice and selective agonists and antagonists. This indicates: 1) the authors need to better incorporate previous literature into their section (not just this paper but many others have very superficial analysis), and 2) provide better critiques on what should be followup/further studies needed. It is not appropriate to say that future studies should perform experiments that are already performed. There are several other examples of this throughout where the studies are very superficially described.

2) The authors state in future perspectives: "processes still need further exploring by constructing the gene-edited animal models, such as MC5R knock out mouse, or invention of highly specific ligands of MC5R." This is very inaccurate as there are knockout mice and there are specific agonists and antagonists. In fact, there is no mention of the phenotype of knockout mice at all. This is important as the phenotype is non-obese and very little impairment in glucose homeostasis. This should be a small section at the beginning, describing studies in KO mice. This does not mean that MC5R is not important in metabolic homeostasis, but this must be rationalized and explained by the authors.

3) How is MC5R being activated endogenously? Indeed, with MC4R in the neurons it is being activated by presynaptic release of alpha-MSH. But what about when MC5R is on tissues like skeletal muscle? What are the endogenous ligands? Does MSH circulate? Is it a local release? This should be described and hypothesized in the metabolic section.

4) the title currently says "hemostasis"

Author Response

In the following review, the authors detail MC5R and its potential role in energy homeostasis. Overall, the first few sections are well written and give a good description of the distribution and pharmacology of MC5R. However, the impact of MC5R on energy homeostasis is superficial, and many statements made already have been examined or performed. It is suggested that the authors re-work that section to incorporate more previous findings and lead to better analysis of the current literature and proposal of future studies.

Thanks for your comments. We have carefully reviewed and summarized the published papers on MCRs, and then we revised the inappropriate statements in the original manuscript. Based on comprehensive understanding of the published papers, we provided new perspectives, as shown in the response to the following questions.

Q1) For example, the authors state in line 233 of pg 9 - " Further study using the Mc5r-/- model or selective ligands for MC5R can help to elucidate the mechanisms of MC5R involvement in glucose uptake in skeletal muscle." However, this is exactly what citation 113 did - they utilized knockout mice and selective agonists and antagonists. This indicates: 1) the authors need to better incorporate previous literature into their section (not just this paper but many others have very superficial analysis), and 2) provide better critiques on what should be follow up/further studies needed. It is not appropriate to say that future studies should perform experiments that are already performed. There are several other examples of this throughout where the studies are very superficially described.

As you suggested, we have carefully read the previous literatures and revised the inappropriate sentence in the manuscript. We deleted the sentence of “using Mc5r-/- model or selective ligands in future” but added the new sentence that “Tissue-specific knockout of the Mc5r will likely yield clues to the functions of the MC5R in different tissues” in future perspective (Page 12, Line 365-366).

We also added Figure 1 in the manuscript to show multiple functions of MC5R in various tissues (Page 2, Line 66-67). A new table, Table 3, comparing the function of MC3R, MC4R, and MC5R in regulation of the energy homeostasis, were added in the revised manuscript (Page 14, Line 374).

Q2) The authors state in future perspectives: "processes still need further exploring by constructing the gene-edited animal models, such as MC5R knock out mouse, or invention of highly specific ligands of MC5R." This is very inaccurate as there are knockout mice and there are specific agonists and antagonists. In fact, there is no mention of the phenotype of knockout mice at all. This is important as the phenotype is non-obese and very little impairment in glucose homeostasis. This should be a small section at the beginning, describing studies in KO mice. This does not mean that MC5R is not important in metabolic homeostasis, but this must be rationalized and explained by the authors.

We have deleted the sentence of “knock out mouse, or invention of highly specific ligands” from the manuscript. Moreover, we comprehensively summarized phenotype of Mc3r, Mc4r, and Mc5r knockout mice in the manuscript (Page 8, Line 207-223).

Q3) How is MC5R being activated endogenously? Indeed, with MC4R in the neurons it is being activated by presynaptic release of alpha-MSH. But what about when MC5R is on tissues like skeletal muscle? What are the endogenous ligands? Does MSH circulate? Is it a local release? This should be described and hypothesized in the metabolic section.

These questions have been answered and addressed in the manuscript as followings (Page 11, Line 314-321).

Q4) the title currently says "hemostasis"

We have revised the “energy homeostasis” by “energy metabolism” for MC5R. “Energy metabolism” can be defined as the processes that underlie food intake, burning the food to release energy, and storing the excess for the time of energy shortage. Therefore, the function of MC5R should be defined as “energy homeostasis”. The current title is: Melanocortin-5 receptor: pharmacology and its regulation of energy metabolism.

Reviewer 2 Report

The manuscript “Regulation of energy hemostasis by melanocortin-5 receptor” aims to review the role of the MC5R in energy homeostasis by regulating lipolysis in adipose tissue and fatty acid oxidation in skeletal muscle.  However, the review suffers from paucity of literature data on: a) expression level of MC5R and MRAPs in human and rodent tissues; b) murine models with overall and tissue-specific deficiency of MC5R.

1) Fig. 4 is quite confusing, are data showing MC1R protein expression levels, as indicated in the figure legend, or instead those of MC5R protein ? it is also confusing the disparity between the human MC5R protein expression level in different tissues as reported in Fig.4 and the MC5R mRNA expression data reported in ref. 84 of table 1.  Humans do not appear to have any expression of MC5R protein in adipose tissue and heart (Fig. 4) and yet have  abundant MC5R mRNA expression in adipose tissue and heart as for ref 84. It is quite difficult to make any point here with the references available so far.

2) "MC5R primarily regulates energy metabolism via adipocyte lipolysis and re-esterification, fatty acid oxidation, and glucose uptake"  Again, this sentence does not go together with expression of MC5R protein in human tissues (Fig.4).

3) Are MRAPs expressed together with MC5R in human and murine tissues? If this is not known, then conclusions on effects of human/murine MRAPs to modulate MC5R pharmacology appear restricted to the non-physiological cell models presented in Table 2, and this point should be made.

3) "alpha-MSH- stimulated MC5R activates HSL and PLIN1 which induce lipolysis by cAMP/PKA signaling pathways, whereas MC5R prevents TAG synthesis by inhibiting the function of acetyl-CoA carboxylase (ACC), an important enzyme in lipogenic process (100)" .  Please add research article reference in addition to to the review in reference 100.

4) "In addition, MC5R in 3T3-L1 adipocytes can inhibit leptin secretion, indirectly regulating food intake and energy expenditure (102)". Please modify this sentence to indicate modulation of energy expenditure by MC5R expressed in adipocytes as a possibility, rather than a conclusion. On that, it does not appear that in Ref 102 there are any animal- based further developments of data obtained from the 3T3-L1 adipocytes, so that any conclusion on energy homeostasis appear as premature.

5) “Activated MC5R triggers cAMP-PKA-AMPK pathway, followed by ACC phosphorylation, which suppresses ACC activity but increases CPT-1 activity, leading to improved FAO (89) (Figure 5).  Please add the pathway to Fig.5.

6) “Future perspectives” Given the limits of the literature on MC5R and MRAP tissue distribution data in human and rodent tissues, it appears to be important to establish precisely the expression level of these factors at the mRNA and protein level.  

Author Response

The manuscript “Regulation of energy hemostasis by melanocortin-5 receptor” aims to review the role of the MC5R in energy homeostasis by regulating lipolysis in adipose tissue and fatty acid oxidation in skeletal muscle.  However, the review suffers from paucity of literature data on: a) expression level of MC5R and MRAPs in human and rodent tissues; b) murine models with overall and tissue-specific deficiency of MC5R.

Thank you for your comments. We have supplemented the data about mRNA expression of MC5R and MRAPs in human tissues in Figure 5 (Page 6, Line 131-134). There are two papers establishing the Mc5rknockout mice in the whole body [1,2]. The phenotype and functional deficiency were also added into the manuscript (Page 8, Line 213-223).

Q1) Fig. 4 is quite confusing, are data showing MC1R protein expression levels, as indicated in the figure legend, or instead those of MC5R protein ? it is also confusing the disparity between the human MC5R protein expression level in different tissues as reported in Fig.4 and the MC5R mRNA expression data reported in ref. 84 of table 1. Humans do not appear to have any expression of MC5R protein in adipose tissue and heart (Fig. 5) and yet have abundant MC5R mRNA expression in adipose tissue and heart as for ref 84. It is quite difficult to make any point here with the references available so far.

Thank you for the suggestion. Figure 5A showed the human MC5R mRNA levels, which has been correct as “Human MC5R mRNA expression” in the legend of Figure 5 (Page 6, Line 132-134).

The assessment of mRNA levels in Human Protein Atlas database is based on the deep-sequencing transcriptomics of the tissues (Figure 5). Human MC5R mRNA (Table 1) distribution in the Québec Family was analyzed by the RT-PCR method. The disparity of human MC5R mRNA distribution between Figure 5 and Table 1 might partially result from the differently detective method. Herein, we added some contents, stating the disparity, in the manuscript (Page 5, Line 122-128).

Q2) "MC5R primarily regulates energy metabolism via adipocyte lipolysis and re-esterification, fatty acid oxidation, and glucose uptake". Again, this sentence does not go together with expression of MC5R protein in human tissues (Fig.5).

Thank you for the suggestion. Human MC5R mRNA did not have expression in adipose tissues in Human Protein Atlas (Figure 5A); however, it showed expression in fat tissues in the Québec Family (Table 1). As we have discussed in the Question 1), the disparity of human MC5R mRNA expression in adipose tissues might partially attribute from the different quantitative methods. In addition, human MC5R mRNA had no expression in adipose tissues at the tissue level in Human Protein Atlas database (Figure 5A); however, it is expressed in the adipocyte and adipose progenitor cell at the cell levels in Human Protein Atlas database (Supplementary Figure 1 in cover letter). These cells are important components of adipose tissues. Therefore, we deduced human MC5R is likely expressed in adipose tissues.

Based on the above explanation, we keep this sentence in the manuscript, but it is a good suggestion which reminds future researchers to get more evidence of MC5R mRNA expression in adipose tissues.

Supplementary Figure 1. MC5R mRNA expression in the subcutaneous adipose cell types based on HPA database (https://www.proteinatlas.org/.)

Q3) Are MRAPs expressed together with MC5R in human and murine tissues? If this is not known, then conclusions on effects of human/murine MRAPs to modulate MC5R pharmacology appear restricted to the non-physiological cell models presented in Table 2, and this point should be made.

We added the following in the manuscript (Page 7-8, Line 193-198):

Co-expression of MC5R and MRAP in the same cells or tissues is the rationale of their interaction. Human Protein Atlas database showed that human MC5R mRNA and MRAP1/MRAP2 are expressed in the same tissues including brain, esophagus, testis, epididymis, skin, and thymus (Figure 5). Similarly, mouse Mc5r and Mrap2 mRNA are expressed in brain, skin, muscle, and adipose. Future research should

systematically investigate the interaction of MC5R and MRAPs in these tissues.

Q4) "a-MSH-stimulated MC5R activates HSL and PLIN1 which induce lipolysis by cAMP/PKA signaling pathways, whereas MC5R prevents TAG synthesis by inhibiting the function of acetyl-CoA carboxylase (ACC), an important enzyme in lipogenic process (100)".  Please add research article reference in addition to the review in reference 100.

The reference titled “α-MSH signalling via melanocortin 5 receptor promotes lipolysis and impairs re-esterification in adipocytes, Biochimica et Biophysica Acta (BBA)-Molecular and Cell Biology of Lipids 1831(7) (2013) 1267-1275.” was inserted into the manuscript (Page 11, Line 287).

Q5) "In addition, MC5R in 3T3-L1 adipocytes can inhibit leptin secretion, indirectly regulating food intake and energy expenditure (102)". Please modify this sentence to indicate modulation of energy expenditure by MC5R expressed in adipocytes as a possibility, rather than a conclusion. On that, it does not appear that in Ref 102 there are any animal- based further developments of data obtained from the 3T3-L1 adipocytes, so that any conclusion on energy homeostasis appear as premature.

We rewrote the sentence in the manuscript as following (Page 8-9, Line 240-242).

"In addition, MC5R in 3T3-L1 adipocytes can inhibit leptin secretion, supporting the possibility that MC5R indirectly regulates food intake and energy expenditure by leptin-melanocortin pathways."

Q6) “Activated MC5R triggers cAMP-PKA-AMPK pathway, followed by ACC phosphorylation, which suppresses ACC activity but increases CPT-1 activity, leading to improved FAO (89) (Figure 5).  Please add the pathway to Fig.5.

The pathway (cAMP-PKA-AMPK-ACC-CPT-1) has been added in Figure 6 in the revised manuscript(Page 10, Line 274).

Q7) “Future perspectives” Given the limits of the literature on MC5R and MRAP tissue distribution data in human and rodent tissues, it appears to be important to establish precisely the expression level of these factors at the mRNA and protein level. 

As you suggested, we have added this point in the “Future Perspectives” as following (Page 12, Line 357-360).

In vitro studies showed that MRAPs can regulate the pharmacology of MC5R in HEK293 or CHO cells, indicating the potential of MRAPs to regulate the function of MC5R. Co-expression of MC5R and MRAP1/MRAP2 in different tissues and the functional regulation of MC5R by MRAPs in a physiological environment need to be studied.

Reviewer 3 Report

The additional Table or Figure and main text, which show the comparison of MC5R and MCR4 or MCR3 in regulation of feeding behavior, energy homeostasis, glucose and lipid metabolism, will be helpful for general readers. 

The authros showed the schematic diagram of MC5R signaling pathways in lipid and glucose metabolism in Figure 5, but the additonal shema, which shows the role of MC5R in the regulation of glucose and lipid metablism in organs including brain, muscle, adipose tissues, and liver, will be also helpful for readers.

Author Response

Q1) The additional Table or Figure and main text, which show the comparison of MC5R and MC4R or MC3R in regulation of feeding behavior, energy homeostasis, glucose and lipid metabolism, will be helpful for general readers. 

Thanks for your suggestion. Table 3 was added to compare the functions of MC5R, MC4R or MC3R in regulation of energy homeostasis (Page 14, Line 374).

Q2) The authors showed the schematic diagram of MC5R signaling pathways in lipid and glucose metabolism in Figure 5, but the additional shema, which shows the role of MC5R in the regulation of glucose and lipid metabolism in organs including brain, muscle, adipose tissues, and liver, will be also helpful for readers.

As you suggested, we have supplemented the tissues associated with MC5R signaling pathways inFigure 6 of the manuscript (Page 10, Line 274). In addition, we have added a new figure listing the functions of MC5R in different tissues or organs in Figure 1 of the manuscript (Page 2, Line 66-67). Figure 1, summarizing the potential function of MC5R in brain, muscle, adipose tissues, liver, spleen, skin, eye, and lung, will be beneficial for readers to understand the overall function of MC5R (Page 2, Line 47-56).

Round 2

Reviewer 1 Report

The authors have done a good job addressing the reviewer's comments. I only have 2 minor concerns:

1) The authors describe that the knockout models have no metabolic phenotype but then detail the effects of MC5R in adipose and skeletal muscle. The authors should address this discrepancy - is the pathway redundant or is there something wrong with the KO mice?

2) The authors have improved the future perspectives section - but there should still be more discussion as to what future studies could elucidate the role of MC5R is in energy metabolism - is there a strong role given the non-phenotype in KO mice? How plausible is pituitary alpha-MSH release as the mediator - do levels change enough to cause differences in binding to MC5R?  Currently the future perspectives is still very superficial.

Author Response

The authors have done a good job addressing the reviewer's comments. I only have 2 minor concerns:

1) The authors describe that the knockout models have no metabolic phenotype but then detail the effects of MC5R in adipose and skeletal muscle. The authors should address this discrepancy - is the pathway redundant or is there something wrong with the KO mice?

Thank you for your comments. The discussion about this question has been added into the manuscript (Page 9, Lines 237-239). We suspect there is also developmental compensation but that is speculation.

2) The authors have improved the future perspectives section-but there should still be more discussion as to what future studies could elucidate the role of MC5R is in energy metabolism - is there a strong role given the non-phenotype in KO mice? How plausible is pituitary alpha-MSH release as the mediator - do levels change enough to cause differences in binding to MC5R?  Currently the future perspectives are still very superficial.

Thank you for your comments. As you suggested, we have added more discussion into manuscript (Page 12).

Reviewer 2 Report

Please indicate da Co-expression of MC5R and MRAP1/MRAP2 in different tissues is to be done at the cellular level.  This is because co-experession of NC4R and MRAP1/2 in a tissue does not necessarily imply co-expression in the same cell.

Author Response

Thank you for your comments. We have revised the sentence “co-expression of MC5R and MRAP1/MRAP2 in different tissues” into “co-expression of MC5R and MRAP1/MRAP2 in the same cells” throughout the manuscript (Page 7, Line; Page 12, Line 339-340).